# Identification and Validation of a Prognostic Signature for Thyroid Cancer Based on Ferroptosis-Related Genes

**DOI:** 10.3390/genes13060997

**Published:** 2022-06-01

**Authors:** Yue Wang, Jing Yang, Shitu Chen, Weibin Wang, Lisong Teng

**Affiliations:** 1Department of Surgical Oncology, the First Affiliated Hospital, School of Medicine, Zhejiang University, Hangzhou 310003, China; wangyue0605@zju.edu.cn (Y.W.); 1319119@zju.edu.cn (S.C.); wbwang@zju.edu.cn (W.W.); 2Institute of Translational Medicine, Zhejiang University, Hangzhou 310020, China; 11918180@zju.edu.cn; 3Women’s Hospital, Zhejiang University School of Medicine, Hangzhou 310006, China

**Keywords:** ferroptosis, thyroid cancer, AKR1C3, prognosis, bioinformatics

## Abstract

Background: Thyroid cancer is the most common endocrine malignancy. Most PTC patients have a good prognosis; however, there are 5–20% of PTC patients with extra-thyroidal invasion, vascular invasion, or distant metastasis who have relatively poor prognoses. The aim of this study is to find new and feasible molecular pathological markers and therapeutic targets for early identification and appropriate management. Methods: The GEO and TCGA databases were used to gather gene expression data and clinical outcomes. Based on gene expression and clinical parameters, we developed a ferroptosis-related gene-based prognostic model and a nomogram. CCK-8, wound-healing, and transwell assays were conducted to explore the proliferation, migration, and invasion abilities of thyroid cancer cells. Results: We found 75 genes associated with ferroptosis that were differentially expressed between normal thyroid tissue and thyroid cancer tissues. The prognostic values of the 75 ferroptosis-related gene expressions were evaluated using the TCGA-THCA dataset, and five (AKR1C3, BID, FBXW7, GPX4, and MAP3K5) of them were of significance. Following that, we chose AKR1C3 as the subject for further investigation. By combining gene expression and clinical parameters, we developed a ferroptosis-related gene-based prognostic model with an area under the curve (AUC) of 0.816, and the nomogram also achieved good predictive efficacy for the three-year survival rate of thyroid cancer patients. Knocking down AKR1C3 enhances thyroid cancer cell proliferation, invasion, and migration abilities. Conclusions: A ferroptosis-related gene-based prognostic model was constructed that provided unique insights into THCA prognosis prediction. In addition, AKR1C3 was found to be a progression promoter in thyroid cancer cell lines.

## 1. Introduction

Thyroid cancer is the most common endocrine malignancy and accounts for approximately 94.5% of all endocrine tumors [1]. The incidence of thyroid cancer is related to region, race, and gender, with the incidence in women being about three times higher than that in men. In recent years, its incidence has been increasing at an annual rate of about 4%, and it has become one of the few cancers with an increasing incidence [2,3,4]. Papillary thyroid carcinoma (PTC), which accounts for more than 80% of the cases, is the most predominant type of thyroid cancer. Most PTC patients have a slow progression, which is highly inert and has a good prognosis. However, there are 5–20% of PTC patients with extra-thyroidal invasion, vascular invasion, or distant metastasis, who have relatively poor prognoses [5,6]. For this group with invasive papillary thyroid cancer with high malignancy, it is particularly important to find new and feasible molecular pathological markers and therapeutic targets for early identification and appropriate management.

Ferroptosis, first proposed by Stockwell et al. in 2012, is a novel form of cell death caused by lipid peroxidation with iron ion dependence [7,8]. Dysregulation of steric function and oncogene regulation in tumor cell lines generate a large accumulation of ROS, leading to a state of high oxidative stress. The rapid growth of tumor cells needs more lipids to support the formation of cell and organelle membranes, and excessive production of lipid ROS causes lethal damage to tumor cells. The ferroptosis caused by lipid ROS may selectively kill tumor cells [7,9]. Currently, scientists have identified a variety of genes and proteins that regulate ferroptosis, including IREB2, TP53, and PIK3CA that serve as drivers to promote ferroptosis, while SQSTM1, MTOR, and STAT3 serve as suppressors to prevent ferroptosis [7,9]. However, it is unknown if these ferroptosis-related genes have an impact on thyroid cancer development and progression.

The goal of this study is to look at the potential value of ferroptosis-related genes as biomarkers in thyroid cancer patients, as well as the key functions of ferroptosis-related genes in thyroid cancer progression.

## 2. Methods and Materials

### 2.1. Microarray Data

The gene expression profiles of the GSE33630, GSE35570, and GSE60542 datasets were downloaded from the GEO database (https://www.ncbi.nlm.nih.gov/geo/ (accessed on 11 January 2022)). ATC and MTC are not included in the scope of this study.

The GSE33630 dataset included 49 PTC patients and 45 cases of normal thyroid, which were all included in our study (ATC (*n* = 11) is excluded from our study). The GSE35570 dataset included 65 PTC patients and 51 cases of normal thyroid. Radiation-induced PTC (*n* = 33) is not within the scope of our study. Thus, we included 32 cancer samples and 51 normal samples. The GSE60542 dataset included 33 PTC samples and 30 normal thyroid samples. LNM samples (*n* = 23), normal lymph node samples (*n* = 4), recurrence samples (*n* = 1), and pleural metastasis samples (*n* = 1) were excluded. Additionally, the RNA sequencing data of THCA were acquired from The Cancer Genome Atlas (TCGA) database (https://www.cancer.gov/tcga/ (accessed on 12 January,2022)).

From the FerrDb database (http://www.zhounan.org/ferrdb/ (accessed on 12 January 2022)), a total of 259 ferroptosis-related genes, including drivers, suppressors, and markers, were collected (Appendix A). Among them, 75 were chosen as candidate genes, and the details are presented in Appendix A. The publication requirements for the GEO and TCGA datasets were followed in this work.

### 2.2. Identification and Functional Study of Differentially Expressed Ferroptosis-Related Genes

The GEO GSE33630, GSE35570, and GSE6054 datasets were used to detect the DEGs between tumor and adjacent normal tissues via the R package “limma” in RStudio (version 3.42.2), with the following cutoff for adjustment: *p*-value < 0.05 and |log2FC| > 1. The R package “ComplexHeatmap” (version 2.2.0) was applied to visualize the degree range of differences in the three datasets. The intersection of candidate genes and DEGs was then used to find ferroptosis-related genes. Metascape Online (https://metascape.org/gp/index.html#/main/step1 (accessed on 24 January 2022)) was used for the functional analysis. The ferroptosis-related genes were put into Metascape to conduct a functional analysis as well as to construct a PPI network. MCODE was performed for further study to reveal the highly connected regions. *p* < 0.05 was used as a cutoff value.

### 2.3. Construction of the Ferroptosis-Related Gene Prognostic Model in Thyroid Cancer

The GEO GSE33630, GSE35570, and GSE60542 datasets, and the TCGA-THCA dataset were used to establish the prediction model. DEGs were obtained from the GEO GSE33630, GSE35570, and GSE60542 datasets. Clinicopathological characteristics and survival status of patients were obtained from the TCGA-THCA dataset. A statistical analysis was conducted to study the discrimination capability of AKR1C3 in thyroid cancer patients. The median value of the original AKR1C3 values was used as the threshold. Patients with AKR1C3 values greater than the threshold were included in the AKR1C3_high group and the remaining patients were in the AKR1C3_low group. The Kaplan–Meier survival curve and the log-rank test were performed to assess differences between the AKR1C3_high and AKR1C3_low groups.

Patients who survived after treatment were classified within a survival group, whereas those who died were classified within a non-survival group. In order to discriminate between the survival and non-survival groups, AKR1C3 and all clinical factors (including pathologic stage, T stage, N stage, extrathyroidal extension, and residual tumor) were normalized. The Z-score normalization was utilized so that all prediction factors had the same scale. A prediction model, named the AKR1C3_clinical model, was constructed using all normalized prediction factors and logistic regression classifiers for predicting survival status in patients with thyroid cancer. A clinical model was also developed based on all normalized clinical factors and logistic regression classifiers. Obviously, the difference between the AKR1C3_clinical model and the clinical model was that the former contained AKR1C3, while the latter did not. We further compared the performances of the proposed AKR1C3_clinical and clinical models. The discrimination performance was quantified by area under the receiver operating characteristic (ROC) curve (AUC), accuracy, precision, recall, F1 score, sensibility, and specificity. Finally, the risk score formula of the optimal model was calculated, and its statistical analysis was explored.

### 2.4. A Nomogram Construction

The risk score of the model was integrated as a prognostic component to evaluate the prediction likelihood of one-, two-, and five-year OS, and a nomogram was constructed to offer the survival probability of a specific event. A calibration curve displaying the three-year OS was constructed to visualize the observed rates against the nomogram-predicted probability. The nomogram and calibration curves were plotted using the R package “rms” (version 6.2-0).

### 2.5. Human Cell Lines

The human anaplastic thyroid cancer cell line 8505c and human papillary thyroid cancer cell line TPC-1 were purchased from the German Collection of Microorganisms and Cell Cultures (DSMZ, Braunschweig, Germany). The cell lines were both maintained in 5% CO_2_ at 38 °C and cultured in RPMI medium (Gibco, Rockville, MD, USA) supplemented with 10% FBS (Gibco).

### 2.6. RNA Interference

The RiboBio Company (Guangzhou, Guangdong, China) provided AKR1C3 small interfering RNAs (siRNA-AKR1C3-1) and non-target small interfering RNA (siRNA-NT). The siAKR1C3 sequence was 5′-GGAACUUUCACCAACAGAUTT-3′. Following the manufacturer’s instructions, transfection was carried out using Lipofectamine 3000 transfection reagent (Invitrogen, Carlsbad, CA, USA) in Opti-MEM medium (Gibco, Rockville, MD, USA). After stable transcription, the cells were collected for the next step of the experiment.

### 2.7. RNA Extraction and Quantitative Real-Time PCR

TRIzol reagent (Invitrogen, Carlsbad, CA, USA) was used to extract total RNA from the cultivated cells, which were then treated with DNase I (Promega Corp, Madison, WI, USA). A high-capacity cDNA synthesis kit (Takara Bio, Inc, kusatsu, shiga, Japan) was used to make cDNA from 2 μg total RNA in 30 μL of reaction buffer, according to the manufacturer’s instructions. The ABI StepOnePlus system (Applied Biosustems^®^, Life Technologies, Shanghai, China) was used to detect gene expression data. The thermal cycling conditions were 95 °C for 1 min, followed by 40 cycles of 95 °C for 10 s and 60 °C 40 s. SYBRGreen I (Takara Bio, Inc., Kusatsu, Shiga, Japan) was used in the RT-qPCR to detect mRNA levels. The expression levels of genes were compared to the expression of the housekeeping gene GAPDH. The following primers were used for the RT-qPCR analysis: GAPDH, 5-ACAACTTTGGTATCGTGGAAGG-3/5-GCCATCACGCCACAGTTTC-3 and AKR1C3, 5′-GGGATCTCAACGAGACAAACG-3′/5′-AAAGGACTGGGTCCTCCAAGA-3′. All of the experiments were carried out in quadruplicate and three cell samples were used each time.

### 2.8. Cell-Counting Kit 8

Cell cytotoxicity was measured using the CCK-8 test (Yeasen Technology, Shanghai, China). To assess drug cytotoxicity, cells were seeded at 3 × 10^3^ cells per well in 96-well plates. After that, cells were treated with various doses of IM for 24 h after achieving 60–70 percent confluence. The cells were given 20 μL of CCK-8 for two hours. At 450 nm, the optical density was determined.

### 2.9. Cell Migration and Invasion Assays

After cultivating transfected and untransfected TPC-1 or 8505c cells to 90% confluence, an artificial ”wound” was generated for the cell migration tests. The distance between cells was measured. The ratio of transfected and untransfected TPC-1 or 8505c cell data was used to calculate migration rates. Transwell^®^ plates were used for invasion tests (Corning, Corning, NY, USA). The cells that infiltrated the lower surface of the filter were fixed and stained with haematoxylin after being seeded onto Matrigel-coated filters. The ratio of the transfected group’s invasion value to that of the untransfected group was used to compute invasion rates.

### 2.10. Statistical Analysis

Quantitative data are presented as the means ± standard deviations (SDs). Pictures were compared via ANOVA, followed by Student’s *t*-tests. *p*-Values less than 0.05 were identified as statistically significant. All statistical analyses were performed using SPSS 24.0. The significance level was set at *p*-values < 0.05.

## 3. Results

### 3.1. Differentially Expressed Ferroptosis-Related Gene Signatures in THCA

The details for the GEO datasets used are listed in Table 1. After a normalization and standardization process, 1040 dysregulated genes were obtained from the GEO GSE33630 dataset with 573 upregulated and 467 downregulated genes (Figure 1A). In general, 2061 DEGs were obtained from the GEO GSE35570 dataset, with 1081 upregulated and 980 downregulated genes (Figure 1B). A total of 928 dysregulated genes were selected from the GEO GSE60542 dataset, including 471 upregulated and 457 downregulated genes (Figure 1C).

Next, the ferroptosis-related genes were downloaded from the FerrDb database (Appendix A). The differentially expressed genes (DEGs) obtained from the GEO datasets were intersected with the ferroptosis gene set to obtain “differentially expressed ferroptosis genes”. As shown in the Venn diagram, 75 ferroptosis-related genes were intersected between four datasets (Figure 1D and Appendix A).

Then, in order to underly the mechanisms of those 75 ferroptosis-related genes in THCA, we performed a functional analysis on Matescape Online. The gene ontology (GO) analysis results revealed that the ferroptosis-related genes were mainly enriched in response to chemical stress, stimulus, and cell death (Figure 1E). The Kyoto Encyclopedia of Gene and Genomes (KEGG) pathway analysis indicated that those genes were mainly enriched in apoptotic, autophagy, and aging signaling pathways. In addition, the protein–protein interaction (PPI) network and MCODE plugin revealed the important modules in these dysregulated genes, including MUC1, CDKN1A, MAPK1, CDKN2A, MAPKB, and EGFR (Figure 1G).

### 3.2. Ferroptosis-Related Genes Predict THCA Prognosis

To explore whether the ferroptosis-related genes were related to the prognosis of thyroid cancer, a univariate COX regression analysis was applied. Based on the TCGA-THCA database, five genes were identified (Figure 2A–E). As shown in Figure 2F, AKR1C3, BID, FBXW7, GPX4, and MAP3K5 were independent prognosis signatures of thyroid cancer.

A total of eight models were constructed by permutations (Appendix A). In four of the seven performance indexes, the AKR1C3_clinical model (Model 3) was better than the other seven models. Therefore, for further study, we chose AKR1C3, which has never been reported to function in the development and progression of thyroid cancer. AKR1C3 is a NADP(H) oxidoreductase that belongs to the aldo-keto reductase superfamily and has been proposed as a therapeutic target for a variety of cancers and endocrine illnesses [13]. The median value of the original AKR1C3 values in 498 patients was 0.9256, which was the threshold we set. The patients were classified into two groups on the basis of the threshold: the AKR1C3_high group and the AKR1C3_low group. The Kaplan–Meier survival curves of the AKR1C3_high and AKR1C3_low groups are plotted in Figure 3A. The blue solid line and orange solid line are the survival curve of the AKR1C3_high group and the AKR1C3_low group, respectively. The light color band represents the 95% confidence interval (CI). In the two Kaplan–Meier survival curves, the survival probability of thyroid cancer patients decreases with an increase in time. The log-rank test shows a significant difference between survival curves of the AKR1C3_high group and the AKR1C3_low group (*p* = 0.021), illustrating that AKR1C3 can be used as one of the prediction factors for evaluating the survival status in patients with thyroid cancer.

AUC, accuracy, precision, recall, F1 score, sensibility, and specificity values of the AKR1C3_clinical model were 0.816, 0.853, 0.990, 0.857, 0.919, 0.857, and 0.750, respectively. For the clinical model, the seven performance indexes were 0.775, 0.799, 0.987, 0.803, 0.886, 0.803, and 0.688, respectively. The performance comparison of the AKR1C3_clinical model and the clinical model is shown in Table 2. The ROC curves of these two models are provided in Figure 3B. We found that all performance indexes of the AKR1C3_clinical model were better than those of the clinical model. Among all patients (survival group, 482 and non-survival group, 16), the number of patients correctly predicted by the two models was 393 (survival patients, 382 and non-survival patients, 11), accounting for 78.92% of the patients and the number of patients who could not be correctly predicted by both models was 68 (survival patients, 64 and non-survival patients, 4), accounting for 13.65% of the patients, demonstrating the effectiveness of our models for predicting the survival status of thyroid cancer patients. The number of patients that the AKR1C3_clinical model correctly predicted but the clinical model could not correctly predict, whether in the survival group or the non-survival group, was greater than the number of patients that the clinical model correctly predicted but the AKR1C3_clinical model could not correctly predict (Appendix A). It shows that the AKR1C3_clinical model has better discrimination capability than the clinical model. The prediction performance of the signature-based risk score for OS was also evaluated using time-dependent ROC curves. The values for the area under the curve were: 0.941 for one-year OS, 0.945 for two-year OS, and 0.795 for five-year OS (Figure 3B,C).

The risk scores of the AKR1C3_clinical model were calculated using the following formula: Risk_score = 4.03 − 0.65 × pathologic_stage − 0.72 × T_stage + 0.15 × N_stage + 0.16 × extrathyroidal_extension − 0.10 × residual_tumor − 0.35 × AKR1C3 (Figure 3D). According to the prediction results of the formula, all patients were divided into a predictive survival group and a predictive non-survival group. The Kaplan–Meier survival curves of the predictive survival and non-survival groups are shown in Figure 3E. A log-rank test was used to evaluate the difference between two survival curves. The *p*-value was 3.34 × 10^−13^ (<0.05), indicating that there was a significant difference between the predictive survival group and the predictive non-survival group when using the AKR1C3_clinical model.

### 3.3. Construction of the Nomogram

Based on the above results, a predictive ferroptosis-related prognostic nomogram was also established. As shown in Figure 3F, this nomogram, which is based on AKR1C3 expression and patient characteristics, including T stage, N stage, pathologic stage, and extrathyroidal extension, is able to predict a thyroid cancer outcome with a satisfying C-index of 0.870 (95% CI 0.838–0.903). As shown in the calibration curve (Figure 3G), the predictive probability is in accordance with the three-year OS.

### 3.4. Downregulation of AKR1C3 Inhibited the Viability of Thyroid Cancer Cell Lines

Through the above experiment we found that patients with high AKR1C3 expression had a worse prognosis, which indicated that AKR1C3 may play a facilitator role in the progression process in thyroid cancer. To further explore the biological significance of AKR1C3 in THCA tumor progression, the 8505c and TPC-1 cells were transfected with siRNA targeting AKR1C3 (siAKR1C3) or negative control siRNA (siNC). As shown in Figure 4B, efficient depletion of AKR1C3 was confirmed by RT-qPCR. The wound-healing assay, cytotoxicity CCK-8 assay, and transwell assay were used to assess the effect of AKR1C3 in thyroid cancer cell proliferation. As shown in Figure 4C–H, downregulation of AKR1C3 significantly inhibited proliferation, invasion, and migration in both cell lines as compared with the control group.

## 4. Discussion

The prognosis of thyroid cancer patients is related to many factors including clinical factors and genes. Previous studies have reported that age, TNM stage, extrathyroidal extension, and lymph node metastasis were considered to be independent risk factors for the prognosis of thyroid cancer patients [14,15]. Genetic alterations such as BRAF and TERT mutations can also affect the survival rate of thyroid cancer patients [16] Ferroptosis, a recently discovered type of programmed cell death, appears to play a critical role in carcinogenesis and cancer therapy efficacy, according to mounting data [17,18]. Prognostic models based on ferroptosis-related genes are becoming a new research hotspot for predicting OS in various malignancies, with prognostic models based on public databases and next generation sequencing (NSG) providing more comprehensive clinical-genetic prognostic value. The predictive usefulness of ferroptosis-related genes for THCA patients’ OS is uncertain, and more research is needed.

In this study, first, we investigated DEGs from the GEO GSE33630, GSE35570, and GSE60542 datasets and compared them to a verified ferroptosis gene collection from the FerrDb database; we discovered 75 ferroptosis-related genes. Then, the 75 ferroptosis-related genes were put through a functional analysis, which revealed that they were linked to cell death, etc. To identify the ferroptosis genes with poor prognosis and create a ferroptosis-based prognostic model, we conducted univariate and multivariate Cox regression analysis.

After that, we collected the ferroptosis-related genes from the FerrDb database. Notably, all five genes we selected had been validated. Sandra Neitemeier et al. [19]. confirmed that mitochondrial transactivation of BID was the ultimate execution step in this oxidative cell death paradigm, linking ferroptosis to mitochondrial damage. Zeng Ye et al. discovered that FBW7 enhanced the lethal effect of gemcitabine in pancreatic cancer by inducing ferroptosis and apoptosis [20]. In previous studies, BID, FBXW7, and GPX4 had already been found to be significant in thyroid cancer [21,22,23], whereas the role of AKR1C3 and MAP3K5 in the development and progression of thyroid cancer has not yet been reported.

Permutations were used to create a total of eight models, as shown in Appendix A. Although Model 8 (which included all clinical parameters as well as five ferroptosis-related genes) had the greatest AUC value, our AKR1C3_clinical model (Model 3) had the best accuracy. The AKR1C3_clinical model (Model 3) outperformed the other seven models in four of the seven performance indexes. Meanwhile, the introduction of one gene signature can reduce cost and have more clinical application value than the introduction of five ferroptosis-related gene signatures. It indicates the superiority of our proposed AKR1C3_clinical model in both experimental performance and clinical cost.

Recently, aldo-keto reductase was found to play a significant role in tumor development and cancer drug resistance [24,25]. As a member of the aldo-keto reductase family 1, AKR1C3 has been shown to be increased in erastin-resistant DU-145 prostate cancer cells in a recent study, suggesting that AKR1C3 might be a suppressor of cell ferroptosis [26]. It was confirmed by the prognostic model and the nomogram that AKR1C3 has an important role in the survival prediction of thyroid cancer patients. To achieve a deeper understanding of the effect that AKR1C3 enrolled, CCK-8, wound healing, and transwell assays were conducted. These findings showed that AKR1C3 downregulation may have an inhibition function in thyroid cancer cells.

In addition to this study, Qian et al. [27] and Lin et al. [28] investigated ferroptosis-related genes in palliative thyroid carcinoma. Different from Qian et al. and Lin et al. who only used the data in the TCGA-THCA database to build a predictive model for OS, we first screened the differential genes from the GEO database, and then validated them in the TCGA database. Choosing data from different database sources improved the validity of our experimental results. Meanwhile, from the perspective of reducing the economic burden of patients, the combination of clinicopathological factors and gene expression in the construction of a prediction model is more valuable for clinical application. We believe our research provides new insights into thyroid cancer treatment.

There are a few limitations in this study. Our risk model was created and verified using public databases; however, additional prospective real-world data are required to substantiate its clinical importance. Second, we only looked at the effect of reduced AKR1C3 expression on biological processes; the specific mechanism in vivo and in vitro needs to be explored further.

## 5. Conclusions

We developed a predictive model based on ferroptosis-related genes that was highly linked to thyroid cancer progression. In addition, we discovered that AKR1C3 is a facilitating factor in thyroid cancer progression.

## Figures and Tables

**Figure 1 genes-13-00997-f001:**
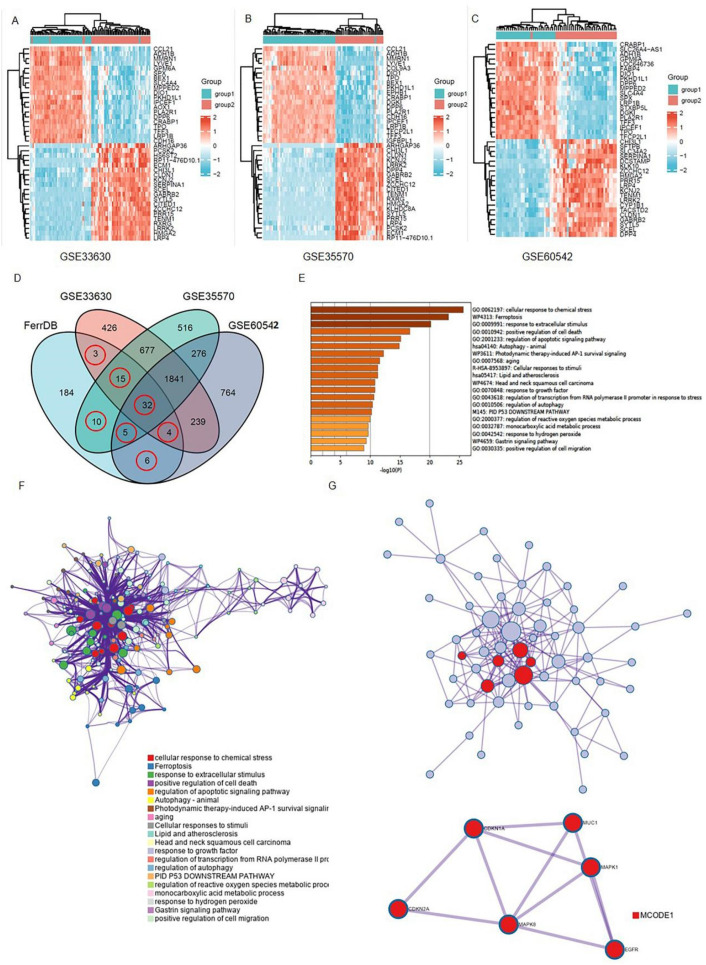
Brief summary of the ferroptosis signals that are differentially expressed in THCA: (**A**–**C**) Ferroptosis-related gene expression profiles in normal and tumor samples in the datasets. The expression of genes was used to group them together. High expression is represented by red, whereas low expression is represented by blue; (**D**) a Venn diagram depicting the dysregulated ferroptosis genes that were found in all four datasets; (**E**,**F**) a graph depicting the GO and KEGG analyses based on Metascape Online, as well as a bar plot and a network depicting the distribution and relationships between the various functions; (**G**) the hub genes in the ferroptosis gene collection are shown in the PPI network and MCODE. Red balls are MCODE1.

**Figure 2 genes-13-00997-f002:**
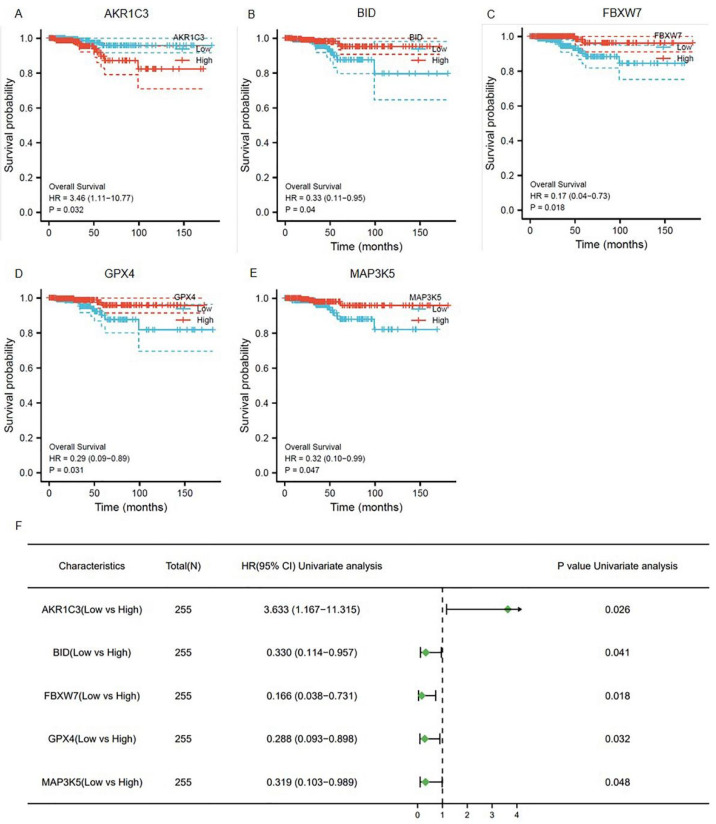
The prognostic ferroptosis signature was plotted using Kaplan–Meier plots and forests plots: (**A**–**E**) Kaplan–Meier plots showing the ferroptosis genes with prognostic values; (**F**) the forest plot showing the results of the univariate Cox regression analyses.

**Figure 3 genes-13-00997-f003:**
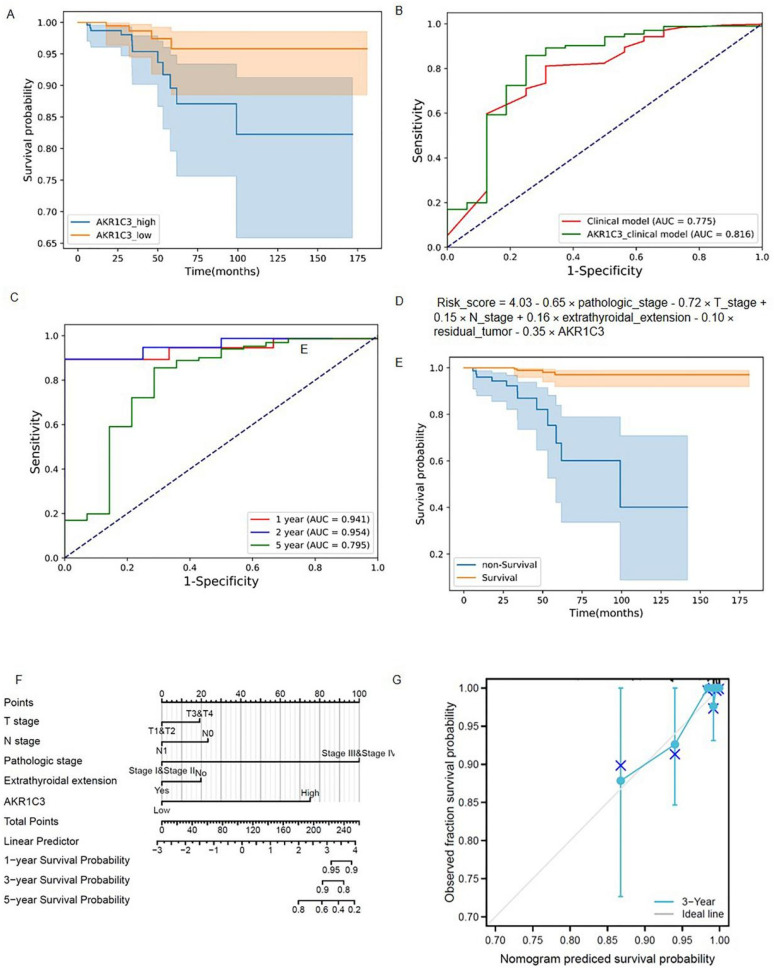
The predictive model construction. (**A**) Kaplan–Meier survival curves of the AKR1C3_high group and AKR1C3_low group; (**B**) ROC curves of the AKR1C3_clinical model and clinical model; (**C**) ROC curves of the AKR1C3_clinical model and clinical model at one, two, and five years; (**D**) risk score of the AKR1C3_clinical model; (**E**) Kapla–Meier survival curves of the predictive survival group and predictive non-survival group using the AKR1C3_clinical model; (**F**) Proposed nomogram to predict 1-,3-, and 5- OS for THCAe; (**G**) calibration curve for the probability of the 3-year OS.

**Figure 4 genes-13-00997-f004:**
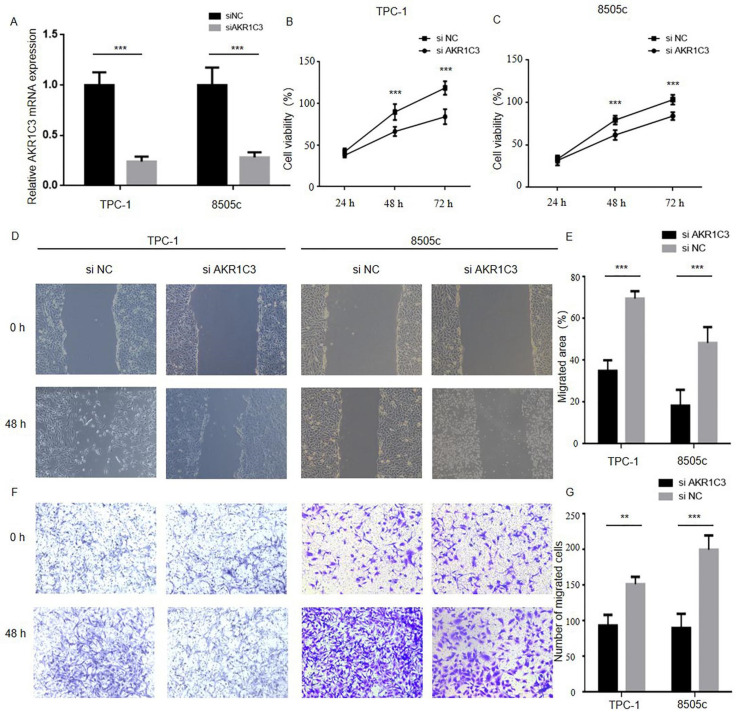
Reducing AKR1C3 expression in thyroid cancer cells inhibits proliferation, migration and invasion: (**A**) TPC-1 and 8505c cell lines were transfected with siNC or siAKR1C3, and the level of PHLPP1 mRNA expression was assessed by RT-qPCR; (**B**,**C**) the CCK-8 assays indicated that knockdown of AKR1C3 inhibits cells proliferation in TPC-1 and 8505c; (**D**–**G**) the wound-healing and transwell asssays suggested that AKR1C3 reduces cell migration and invasion in TPC-1 and 8505c (** *p* < 0.01, *** *p* < 0.001).

**Table 1 genes-13-00997-t001:** The information of datasets from the GEO database.

References	Accession Number	Samples		Year
		Tumor	Normal	
Tomas et al. [10]	GSE33630	49	45	2012
Jarzab et al. [11]	GSE35570	32	51	2015
Tarabichi et al. [12]	GSE60542	33	30	2015

**Table 2 genes-13-00997-t002:** Performances of AKR1C3_clinical model and Clinical model.

	AKR1C3_Clinical Model	Clinical Model
AUC	0.816	0.775
Accuracy	0.853	0.799
Precision	0.990	0.987
Recall	0.857	0.803
F1 score	0.919	0.886
Sensibility	0.857	0.803
Specificity	0.750	0.688

## Data Availability

The datasets supporting our researches are presented in the article.

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
