# Peer review of "Identification and Validation of a Prognostic Signature for Thyroid Cancer Based on Ferroptosis-Related Genes"

_genes, 2022, doi:10.3390/genes13060997_

Round 1

Reviewer 1 Report

The authors developed a predictive model based on ferroptosis-related genes  which they link to overall survival.  The manuscript lacks coherence and not written for brevity.  The methods although are good, have NOT been explained well.

Certain aspects of introduction matches to the text in public domain 

The figures are of not high resolution 

The discussions and conclusions do NOT make coalesce with OS plots. 

I corrected a few things in the pdf.  please find attached

Author Response

Dear Editors and Reviewers:

Thank you for your letter and for the reviewers’ comments concerning out manuscript entitled ‘Identification and Validation of a Prognostic Signature for Thyroid Cancer Based on Ferroptosis-Related Genes’ (ID: genes-1697920). Those comments are all valuable and very helpful for revising and improving our paper, as well as the important guiding significance to our researches. We have studied comments carefully and have made correction which we hope meet with approval. Revised portion are marked in blue in the paper. The main correction in the paper and the responds to the reviewer’s comments are as following:

Reviewer 1#

Response to comment:The authors developed a predictive model based on ferroptosis-related genes which they link to overall survival. The manuscript lacks coherence and not written for brevity. The methods although are good, have NOT been explained well.

Certain aspects of introduction matches to the text in public domain

The figures are of not high resolution

The discussions and conclusions do NOT make coalesce with OS plots.

I corrected a few things in the pdf.  please find attached

Response: We appreciate the constructive comment and careful revision from the referee. We have added content to the Materials and Methods and Discussion sections, which we hope will clearly demonstrate the process and results of our experiments. Meanwhile, clear tables and figures will be uploaded to the system that

Special thanks to reviewers for your good comments.

We tried our best to improve the manuscript and made some changes in the manuscript. We appreciate for Editors/ Reviewers’ warm work earnestly, and hope that the correction will meet with approval.

Once again, thank you very much for your comments and suggestions.

With best regards

Yours sincerely

Yue Wang

Reviewer 2 Report

“Identification and Validation of a Prognostic Signature for Thyroid Cancer Based on Ferroptosis-Related Genes” by Yue Wang et al. They figured out that AKR1C3, a ferroptosis-related gene, could be an oncogenic factor in thyroid cancer. There are several concerns in this study that should be pointed out.

Major

  1. In the introduction: the authors should introduce the role of ferroptosis-related genes in other cancers.
  2. In the materials and methods:
    1. Line 62: please explain “ATC and PTC is not included in the scope of this study.”. This study is focused on PTC?
    2. About the cell lines: please give more information about the cells.
    3. Please describe the origin of your material like DNase I (Promega Corp, USA) for all.
    4. Please describe the protocol for siRNA transfection as well as the details of siRNA- AKR1C3
  3. In the results
    1. Please explain how to figure out five genes (AKR1C3, BID, FBXW7, GPX4, MAP3K5) among overlapped genes. Line 192
    2. As shown in Figure 2.A, high expression of AKR1C3 was associated with poor prognosis. Meanwhile, the invitro experiment revealed that inhibition of AKR1C3 was induced the cell proliferation, migration, and invasion. Please explain these issues.
  4. In the discussion, please discuss more about your results.
    1. The authors mention that “Although Qian et al. and Lin et al. identified ferroptosis-related genes and used the TCGA database to build a predictive model for OS, our methodologies were different” please go into details. Line 305-306
  5. There are many grammatical errors. In addition, there are strange sentences and wordings. The manuscript needs to be thoroughly revised. It is strongly recommended that the manuscript is edited by a native speaker.

Author Response

Dear Editors and Reviewers:

Thank you for your letter and for the reviewers’ comments concerning out manuscript entitled ‘Identification and Validation of a Prognostic Signature for Thyroid Cancer Based on Ferroptosis-Related Genes’ (ID: genes-1697920). Those comments are all valuable and very helpful for revising and improving our paper, as well as the important guiding significance to our researches. We have studied comments carefully and have made correction which we hope meet with approval. Revised portion are marked in blue in the paper. The main correction in the paper and the responds to the reviewer’s comments are as following:

Reponds to the reviewer’s comments:

Reviewer 2#

  1. Response to comment: In the materials and methods:

Line 62: please explain “ATC and PTC is not included in the scope of this study.”. This study is focused on PTC?

Response:We appreciate the constructive comment from the referee. We are very sorry for the writing errors and have made changes. The correct one would be.”ATC and MTC is not incuded in the scope of this study”.

  1. Response to comment: About the cell lines: please give more information about the cells.

Please describe the origin of your material like DNase I (Promega Corp, USA) for all.

Response: We appreciate the constructive comment from the referee. Human anaplastic thyroid cancer cell line 8505c and human papillary thyroid cancer cell line TPC-1 were purchased from German Collection of Microorganisms and Cell Cultures (DSMZ). And these infromation  have been added to the Materials and Methods section.

  1. Response to comment:Please describe the protocol for siRNA transfection as well as the details of siRNA- AKR1C3

Response: We appreciate the constructive comment from the referee. We have added to this section as detailed in the Methods and Materials section 2.6.

In the results

  1. Response to comment:Please explain how to figure out five genes (AKR1C3, BID, FBXW7, GPX4, MAP3K5) among overlapped genes. Line 192

Response: We appreciate the reviewers' questions. In the first part, we obtained 75 ferroptosis-related DEGs. Using univariate COX regression analysis, these genes’ expression and patient survival condition were combined analysed in the TCGA-THCA database. Only these 5 genes of these 75 had a P-value < 0.05. suggesting that they are independent risk factors for the survival of PTC patients.

  1. Response to comment:

As shown in Figure 2.A, high expression of AKR1C3 was associated with poor prognosis. Meanwhile, the invitro experiment revealed that inhibition of AKR1C3 was induced the cell proliferation, migration, and invasion. Please explain these issues.

Response: We appreciate the constructive comment from the referee. In the course of our experiments, we found that patients with relatively low AKR1C3 expression had better OS. However, its expression in thyroid cancer tumor tissues was lower than in normal tissues, which may be due to its different roles in the development and progression of thyroid cancer. And further cytological experiments verified that knocking down the expression of AKR1C3 in thyroid cancer cells could inhibit the proliferation, invasion and migration ability of the cells.

  1. Response to comment:In the discussion, please discuss more about your results.And comment:The authors mention that “Although Qian et al. and Lin et al. identified ferroptosis-related genes and used the TCGA database to build a predictive model for OS, our methodologies were different” please go into details. Line 305-306

Response: We appreciate the constructive comment from the referee. We have expanded this section. “Different from  Qian et al. and Lin et al. only used the data in TCGA-THCA database to build a predictive model for OS, we first screened the differential genes from the GEO database and then validated them in the TCGA database. Choosing data from different database sources will make our experimental results more credible.” (Line 350-354)

  1. Response to comment:There are many grammatical errors. In addition, there are strange sentences and wordings. The manuscript needs to be thoroughly revised. It is strongly recommended that the manuscript is edited by a native speaker.

Response:We appreciate the constructive comment from the referee. We have modified some of these statements and will use the language editing service provided by MDPI to do so if necessary.

Special thanks to reviewers for your good comments.

We tried our best to improve the manuscript and made some changes in the manuscript. We appreciate for Editors/ Reviewers’ warm work earnestly, and hope that the correction will meet with approval.

Once again, thank you very much for your comments and suggestions.

With best regards

Yours sincerely

Yue Wang

Round 2

Reviewer 1 Report

I am well satisfied with all the changes rendered by the authors.   The conclusions could be extended to a fe wmore sentences, just starting with a rationale and justification.

Thank you  

Author Response

We thank the reviewers for their comments and suggestions.

Best wishes.

Reviewer 2 Report

There are still some defects in the manuscript and major concerns in this study that must be clarified.

Major

  1. The most significant issue of this manuscript is that it still has many typos, punctuations, and grammar errors, despite the fact that the authors stated, “We have modified some of these statements”. Examples: “pridiction” line 82, “by Student's ttests.” Line 163, “..” line 269, “mogration" line 284, “durg resistance” line 333, “  ”, etc.
  2. In the results
    1. “As shown in Figure 2F, AKR1C3, BID, FBXW7, GPX4, MAP3K5 were independent prognosis signatures of thyroid cancer”, the multivariate analysis should be applied for analysis the relationship between clinical features and these ferroptosis-Related Genes to make a conclusion like that. Line 201-202
    2. “And through the above experiment we found that patients with low AKR1C3 expression had a worse prognosis, which indicated that AKR1C3 may play a different role in the development and progression process in thyroid cancer”. Line 275-278. The authors may be conflicted with previous data, as shown in Fig 2.A high AKR1C3 expression has a poor prognosis than low AKR1C3 expression in PTC patients.
  3. In the discussion,
    1. Line 296-299 please cite the literature used in your discussion.
    2. Line 324-330 are similar to Line 207-210
    3. “AKR1C3 downregulation may have a inhibition function in thyroid cancer cells, although the exact mechanism has to be investigated further”. Line 339-340, please carefully discuss your results, as your data knockdown of AKR1C3 showed inhibited cell growth at 48h. The invasion and migration of siRNA-ARK1C3 transfected cells were lower than siRNA-negative control cells at 48h which may be affected by the cell growth.

Author Response

Dear Editors and Reviewers:

Thank you for your letter and for the reviewers’ comments concerning out manuscript entitled ‘Identification and Validation of a Prognostic Signature for Thyroid Cancer Based on Ferroptosis-Related Genes’ (ID: genes-1697920). Those comments are all valuable and very helpful for revising and improving our paper, as well as the important guiding significance to our researches. We have studied comments carefully and have made correction which we hope meet with approval. The main correction in the paper and the responds to the reviewer’s comments are as following:

  1. There are still some defects in the manuscript and major concerns in this study that must be clarified.

Response to comment: The most significant issue of this manuscript is that it still has many typos, punctuations, and grammar errors, despite the fact that the authors stated, “We have modified some of these statements”. Examples: “pridiction” line 82, “by Student's ttests.” Line 163, “..” line 269, “mogration" line 284, “durg resistance” line 333, “  ”, etc.

Response: We appreciate the reminder from the referee. We have fixed these errors. Also we have asked the editor of MDPI to make corrections, and hopefully already these errors have been corrected.

  1. In the results

(1) “As shown in Figure 2F, AKR1C3, BID, FBXW7, GPX4, MAP3K5 were independent prognosis signatures of thyroid cancer”, the multivariate analysis should be applied for analysis the relationship between clinical features and these ferroptosis-Related Genes to make a conclusion like that. Line 201-202

Response: Thanks. As shown in the section of 2. To explore whether these ferroptisis-related genes are related to prognosis of thyroid cancer, univariate COX regression analysis was applied. However only 5 (including AKR1C3, BID, FBXW7, GPX4, and MAP3K5) out of 75 genes’ P value were of significant. Therefore, these genes were selected for further research in this manuscript.

(2) “And through the above experiment we found that patients with low AKR1C3 expression had a worse prognosis, which indicated that AKR1C3 may play a different role in the development and progression process in thyroid cancer”. Line 275-278. The authors may be conflicted with previous data, as shown in Fig 2.A high AKR1C3 expression has a poor prognosis than low AKR1C3 expression in PTC patients.

Response: Thanks to the reviewers for the reminder. We apologize for writing errors here. As shown in Figure 3A, the overall survival rate of patients with high AKR1C3 expression was lower. This is consistent with the results of our cellular experiments. Since the main purpose of the Result 4 is to verify the role of AKR1C3 in tumor progression, in this revision we deleted the original Figure 4A.

  1. In the discussion,

(1)Line 296-299 please cite the literature used in your discussion.

(2)Line 324-330 are similar to Line 207-210

(3)“AKR1C3 downregulation may have a inhibition function in thyroid cancer cells, although the exact mechanism has to be investigated further”. Line 339-340, please carefully discuss your results, as your data knockdown of AKR1C3 showed inhibited cell growth at 48h. The invasion and migration of siRNA-ARK1C3 transfected cells were lower than siRNA-negative control cells at 48h which may be affected by the cell growth.

Response: We appreciate the constructive comment from the referee. We have revised the content accordingly. As for the third question, our intention was to refer to the lack of studies on the specific pathways or targets of AKR1C3 that promote thyroid cancer progression in this paper. The second half of the sentence was a bit redundant, and we removed it in this revision.

Special thanks to reviewers for your good comments.

We tried our best to improve the manuscript and made some changes in the manuscript. We appreciate for Editors/ Reviewers’ warm work earnestly, and hope that the correction will meet with approval.

Once again, thank you very much for your comments and suggestions.

With best regards

Yours sincerely
